# Phenotypic and Molecular Characteristics of the MDR Efflux Pump Gene-Carrying *Stenotrophomonas maltophilia* Strains Isolated in Warsaw, Poland

**DOI:** 10.3390/biology11010105

**Published:** 2022-01-10

**Authors:** Olga M. Zając, Stefan Tyski, Agnieszka E. Laudy

**Affiliations:** 1Department of Pharmaceutical Microbiology, Medical University of Warsaw, 02091 Warsaw, Poland; olga.zajac@wum.edu.pl (O.M.Z.); s.tyski@nil.gov.pl (S.T.); 2Department of Antibiotics and Microbiology, National Medicines Institute, 02091 Warsaw, Poland

**Keywords:** antibiotic susceptibility, efflux systems, molecular typing, MLST, PFGE, non-fermentative rods

## Abstract

**Simple Summary:**

Nosocomial infections caused by *Stenotrophomonas maltophilia* have been increasing worldwide. These bacteria are intrinsically resistant to most antibiotics. The underestimated resistance mechanism of Gram-negative rods is an overexpression of multidrug-resistant (MDR) efflux pumps. The aim of this study was to analyze the genetic diversity of isolates derived from various clinical materials, including blood, and the prevalence of MDR efflux pump genes and susceptibility profiles to the anti-*S. maltophilia* drugs. The research was conducted on 94 *S. maltophilia* isolates derived from hospitalized patients and outpatients in Warsaw, Poland. All isolates were susceptible to trimethoprim-sulfamethoxazole and minocycline, while 44/94 isolates demonstrated reduction in susceptibility to levofloxacin. A large genetic variation was observed among these isolates. However, a clonal relationship was revealed among two groups of bloodstream isolates from one hospital ward: (1) nine isolates, (2) six isolates. Moreover, the presence of genes encoding ten different efflux pumps from the resistance-nodulation-division family and the ATP-binding cassette family was shown in the majority of the 94 isolates. The obtained knowledge about the prevalence of efflux pump genes in clinical *S. maltophilia* strains makes it possible to predict the scale of the risk of resistance emergence in strains as a result of gene overexpression.

**Abstract:**

An increase of nosocomial infections caused by *Stenotrophomonas maltophilia* strains has recently been observed all over the world. The isolation of these bacteria from the blood is of particular concern. In this study we performed the phenotypic and genotypic characterization of 94 *S. maltophilia* isolates, including isolates from patients hospitalized in a tertiary Warsaw hospital (*n* = 79) and from outpatients (*n* = 15). All isolates were found to be susceptible to trimethoprim-sulfamethoxazole and minocycline, while 44/94 isolates demonstrated a reduction in susceptibility to levofloxacin. A large genetic variation was observed among the isolates tested by pulsed-field gel electrophoresis. A clonal relationship with 100% similarity was observed between isolates within two sub-pulsotypes: the first included nine bloodstream isolates and the second involved six. Multilocus sequence typing showed two new sequence types (ST498 and ST499) deposited in public databases for molecular typing. Moreover, the presence of genes encoding ten different efflux pumps from the resistance-nodulation-division family and the ATP-binding cassette family was shown in the majority of the 94 isolates. The obtained knowledge about the prevalence of efflux pump genes in clinical *S. maltophilia* strains makes it possible to predict the scale of the risk of resistance emergence in strains as a result of gene overexpression.

## 1. Introduction

*Stenotrophomonas maltophilia* is one of the most frequently isolated non-fermentative Gram-negative bacteria responsible for nosocomial infections [1,2,3]. This opportunistic pathogen is dangerous for patients with coexisting diseases like acquired immunodeficiency syndrome (AIDS), cystic fibrosis, or cancer (particularly lung cancer), as well as patients with other immunodeficiencies, immunosuppressive therapy, mechanical ventilation, catheters, and those hospitalized for long periods. The critically ill patients from intensive care units, especially after broad-spectrum antibiotic therapy, are most exposed to infections. *S. maltophilia* strains mostly cause nosocomial respiratory tract infections (pneumonia), which can be associated with bloodstream infections. These bacteria can cause many other severe infections, including respiratory infections associated with acute exacerbations of chronic obstructive pulmonary disease, eye infections (endophthalmitis, keratitis, scleritis), skin and soft tissue infections, meningitis, endocarditis, and biliary sepsis. Infections caused by *S. maltophilia* are characterized by a high mortality rate, up to 69% in patients with bacteremia [3]. Although it is mainly a nosocomial pathogen, community-acquired infections are increasingly observed. Infections caused by this bacterium occur in both adults and children [1,2]. Infection occurs most often through direct contact with the aerosol created by a person with pneumonia. Other possible routes of transmission involve the hands of healthcare workers, dental unit suction tubing, contaminated endoscopes, or tap water. *S. maltophilia* strains, like most non-fermentative Gram-negative bacteria, are common in wet or moist environments (e.g., in taps or washbasin siphons and their surroundings) [2].

*S. maltophilia* infections are difficult to treat because they are intrinsically resistant to a majority of antibiotics and chemotherapeutic agents. According to the Clinical and Laboratory Standards Institute (CLSI), recommendations for Group A antimicrobial agents which are appropriate for use in the routine treatment of *S. maltophilia* infection includes only the three following antibiotics: trimethoprim-sulfamethoxazole, levofloxacin, and minocycline. Several mechanisms contribute to *S. maltophilia* resistance, among them: β-lactamases production, the activity of multidrug-resistant (MDR) efflux pumps, the excretion of other enzymes modifying antibiotic structures, and the production of proteins that protect the drug targets [2,4].

One underestimated resistance mechanism of Gram-negative rods, including *S. maltophilia* strains, is an overexpression of MDR efflux pumps. Efflux pumps described so far in Gram-negative bacteria can be classified into five families, called superfamilies, as follows: the ABC (ATP-binding cassette) family, the RND (resistance-nodulation-division) family, the MFS (major facilitator superfamily), the SMR (small multidrug resistance) family, and the MATE (multidrug and toxic compound extrusion) family. Of these families, mainly RND efflux systems contribute to antimicrobial resistance. RND efflux systems are able to remove antimicrobial agents belonging to different classes of antibiotics, chemotherapeutic agents, and disinfectants from bacterial cells [4]. These systems are tripartite complexes consisting of inner membrane proteins (IMPs), outer membrane proteins (OMPs), and membrane fusion proteins (MFPs) located in the periplasm combining the IMP and the OMP subunits. These efflux pumps are activated by a proton motive force to extrude agents into the extracellular environment. The RND efflux systems are encoded by genes organized in operons located in bacterial chromosomes. Only two of all known efflux pumps of Gram-negative rods are coded by genes located in conjugational plasmids, that is, the OqxAB and QepA efflux pumps reported among *Enterobacteriaceae* strains.

In contrast to *Enterobacteriaceae* and *Pseudomonas aeruginosa*, the MDR efflux pumps have been relatively sparsely investigated in *S. maltophilia* strains. This applies to both their prevalence among clinical and environmental strains and their role in the resistance of these bacilli. Until recently, it was thought that 12 MDR efflux systems occur in *S. maltophilia*. These systems belong to three different families of MDR efflux pumps: the RND family (SmeDEF [5,6,7], SmeABC [8], SmeIJK [9], SmeYZ [9], SmeVWX [10], SmeOP [11], SmeMN [12], and SmeGH [12]); the ABC family (SmrA [13] and Ma-cABCsm [14]); the MFS family (EmrCABsm) [15]; and the FuaABC pump [16], which is not classified. Moreover, the identification of the next two new MDR efflux pumps from the ABC family, SmaCDEF and SmaAB, in *S. maltophilia* was published in 2021 [17]. There is currently ongoing research on the occurrence of these MDR efflux systems among *S. maltophilia* isolated from humans and animals.

Recent investigations have revealed the high genetic diversity among *S. maltophilia* strains isolated in different parts of the world [18,19]. Molecular methods are used to provide evidence of epidemiological relationships between isolates. These methods are also an important tool in the investigation of the spread of *S. maltophilia* infections all over the world. Rizek et al. sequenced and analyzed the whole genomes of four clinical isolates of *S. maltophilia*, and for the presence of genes encoding the efflux pump systems [20]. The following MDR efflux pump genes have been identified: *smeABC* (in two out of four isolates), *smeDEF* (in three out of four isolates), *smeZ* (in three out of four), *smrA* (in one out of four), and *macB* (in all isolates), among others. It is therefore possible that there are no genes encoding the different pumps in the tested isolates. On the other hand, the presence of efflux-system-coding operons is not synonymous with the resistance of such a strain to antibiotics that are substrates of the MDR pumps. Only the overexpression of efflux pump genes causes resistance or decreased susceptibility of a strain to antibiotics. So, if most clinical *S. maltophilia* strains possess the genes listed above, the scale of the existing danger of the emergence of resistance, even MDR, is huge.

The aim of this study was to analyze the genetic diversity of isolates derived from various clinical materials, including blood, and the prevalence of genes encoding MDR efflux pumps and susceptibility profiles to the main three anti-*S. maltophilia* drugs. The research was conducted on a collection of 94 *S. maltophilia* clinical isolates obtained from hospitalized patients and outpatients in Warsaw, Poland.

## 2. Materials and Methods

### 2.1. Bacterial Strains

The research was conducted on a collection of 94 non-duplicate *S. maltophilia* clinical isolates obtained from adult patients hospitalized in one tertiary hospital in Warsaw (*n* = 79) and from outpatients (*n* = 15). Isolates were derived between January 2010 and October 2013 from blood samples (27), bronchial secretions (17), anus swabs (14), wound swabs (8), urine samples (6), sputum (4), drain swabs (3), stoma swabs (2), oral cavity swabs (2), eye swabs (2), gastrostomy swabs (2), ear swabs (2), peritoneal fluid, a bile sample, fluid from the pleural cavity, a vagina swab, and a nose swab. Biochemical identification of *S. maltophilia* isolates was performed by the Vitek-2 Compact system (bioMérieux, Mercy l’Etoile, France). All isolates were stored in Luria Bertani broth (BioMaxima SA, Lublin, Poland) with 20% glycerol at −80 °C until analysis. Two reference strains of *S. maltophilia* ATCC 13,637 and *S. maltophilia* ATCC 12,714 were also included in this study.

### 2.2. Antimicrobial Susceptibility Testing

The susceptibility of clinical isolates to levofloxacin, minocycline, and trimethoprim-sulfamethoxazole was examined by the disc-diffusion test, according to the Clinical and Laboratory Standards Institute (CLSI) recommendations [21], and by the minimum inhibitory concentration (MIC) determination using Etests (Liofilchem srl, Roseto deli Abruzzi, Italy) [22]. Both assays were determined on Mueller-Hinton II agar medium (Becton, Dickinson and Company, Franklin Lakes, NJ, USA). The results of the susceptibility of the isolates were evaluated after incubation at 35 °C for 18 h and interpreted according to the CLSI criteria [23]. *Escherichia coli* ATCC 25,922 was used as a reference strain for quality control in the antimicrobial susceptibility testing.

### 2.3. Detection of MDR Efflux Pump Genes

The molecular detection of genes encoding the MDR efflux pumps from the RND family (SmeDEF, SmeABC, SmeIJK, SmeYZ, SmeOP, SmeGH, SmeMN, SmeVWX) and the ABC family (SmrA, MacABCsm) was performed by polymerase chain reaction (PCR). The total DNA of the clinical isolates was extracted using a Genomic Mini Kit (A&A Biotechnology, Gdynia, Poland). The PCR reactions were performed using Maxima Hot Start Taq DNA polymerase (Thermo Scientific, Thermo Fisher Scientific, Waltham, MA, USA) with the following amplification parameters: 95 °C for 4 min, followed by 25 cycles of 30 s at 95 °C, 30 s at 58 °C, 59 °C or 63 °C (annealing temperature for each pair of primers is described in Table 1), 60 s at 72 °C, and a final extension for 5 min at 72 °C. The sequences of the primers were derived from references or designed for this project based on gene sequences available in GenBank National Center for Biotechnology Information (NCBI) (https://www.ncbi.nlm.nih.gov/, accessed on 1 February 2018). Primers used in this study are listed in Table 1. The reference *S. maltophilia* ATCC 13,637 strain harboring the MDR efflux pump genes was used. By sequencing the PCR reaction products obtained for the reference strain with primers listed in Table 1, the correct use of these primers for the detection of the specific pumps encoding genes was confirmed.

### 2.4. Pulsed-Field Gel Electrophoresis (PFGE)

All 94 *S. maltophilia* isolates were typed by PFGE according to a protocol published by Jumaa et al. [24] with modifications. An overnight culture of bacteria with a density of about 10^8^ CFU/mL was suspended in 150 µL of cell suspension buffer and mixed with 20 µL of 20 mg/mL Proteinase K (Promega GmbH, Walldorf, Germany) and 170 µL of 1.5% low-melting agarose (SeaKem Gold Agarose Lonza, Basel, Switzerland) to form a plug with bacteria cells. The total DNA in agarose plugs was obtained by material lysis in 2.5 mL lysis buffer supplemented with 20 µL of 20 mg/mL Proteinase K (Promega GmbH, Walldorf, Germany) and 7.5 µL of 10 mg/mL RNase (Sigma, St. Louis, MO, USA) for 2.5 h at 55 °C. Plugs were washed and digested with 15U of *XbaI* restriction enzyme (Thermo Scientific, Thermo Fisher Scientific, Waltham, MA, USA) for 3 h at 37 °C. Electrophoresis was performed on the CHEF DR II Variable Angle System (Bio-Rad, Hercules, CA, USA). The electrophoresis conditions were as follows: total run time 20 h (the first-block switch time was 1 to 12 s for 13 h, and the second-block switch time was 5 to 35 s for 7 h), voltage 6 V/cm, switch angle 120°, temperature 14 °C. The total DNA of *Salmonella* serotype Braenderup strain (H9812) digested with *XbaI* enzyme (ABO, Gdańsk, Poland) was used as the DNA molecular-weight marker. The obtained PFGE profiles were analyzed using GelCompare II software (Applied Maths, Sint-Martens-Latem, Belgium), using the Dice coefficient and clustering by UPGMA with 1% tolerance. According to Tenover et al. [25] the isolates were clustered in the PFGE pulsotypes (PTs). Isolates with banding pattern similarity over 80% were considered to be related.

### 2.5. Multilocus Sequence Typing (MLST)

The total DNA of selected clinical isolates was extracted using a Genomic Mini Kit (A&A Biotechnology, Gdynia, Poland). The methodology of MLST typing was carried out according to the Kaiser et al. [26] protocol. The sequences of the following seven highly conserved housekeeping genes were analyzed: *atpD, gapA, guaA, mutM, nuoD, ppsA,* and *recA*. The sequences of these primers are included in *S. maltophilia* MLST database (https://pubmlst.org/organisms/stenotrophomonas-maltophilia/primers, accessed on 1 February 2020). Sequencing of the obtained DNA templates was carried out in The Laboratory of DNA Sequencing and Oligonucleotides Synthesis, Institute of Biochemistry and Biophysics, Polish Academy of Science in Warsaw, Poland. The received DNA sequences were analyzed using Vector NTI Advance 11 software (Invitrogen, Thermo Fisher Scientific, Waltham, MA, USA) and compared with *S. maltophilia* MLST database (https://pubmlst.org/organisms/stenotrophomonas-maltophilia/, accessed on 1 February 2020). A combination of the allelic sequences of all seven genes enabled the definition of the sequence type (ST) for each isolate. The new allelic sequences of housekeeping genes and new STs were submitted to the *S. maltophilia* MLST database (https://pubmlst.org/organisms/stenotrophomonas-maltophilia/, accessed on 19 June 2020).

## 3. Results

### 3.1. Susceptibility Profiles of the Isolates

Table 2 shows the susceptibility patterns of 94 *S. maltophilia* isolates for three antibacterial agents. All isolates were susceptible to minocycline and trimethoprim-sulfamethoxazole, regardless of the assay used. Moreover, a minority of the studied isolates were nonsusceptible to levofloxacin (44 isolates in the Etest method and 8 isolates in the disc-diffusion method). The highest level of resistance to levofloxacin (MIC = 16 mg/L) was obtained only for isolates 9/2010, 30/2011, and 41/2011. The remaining four out of seven levofloxacin-resistant isolates showed an MIC value of 8 mg/L.

### 3.2. Occurrence of Genes Encoding the MDR Efflux Systems

The PCR analysis of the distribution of MDR efflux system genes among the tested isolates revealed the presence of the *smeD*, *smeN*, *smeH*, and *macB* genes in all isolates, and the following genes encoding other efflux pumps: *smeW* and *smrA* in 93 out of the 94 isolates, *smeP* in 90 isolates, and *smeZ* in 89. However, when searching for the SmeIJK and SmeABC efflux systems, the *smeK* and *smeB* genes were detected in only 69 out of 94 studied isolates. In the case of isolates with a negative result, the *smeI* and *smeA* genes were amplified in the second stage. We found that 17 out of 25 *smeK*-negative isolates harbored the *smeI* gene. On the other hand, the *smeA* gene was only detected among 3 out of 25 *smeB*-negative isolates. The occurrence of MDR efflux systems in the studied isolates of *S. maltophilia* is shown in Table 3, and the presence of the particular genes encoding these systems is given in Appendix A.

Among the 94 collected clinical isolates of *S. maltophilia*, PCR analysis showed the presence of 10 different MDR efflux systems—8 from the RND family and 2 from the ABC family—in 63 isolates. Meanwhile, 31 isolates showed the lack of one (21 isolates) or two (10 isolates) MDR efflux systems. In most isolates lacking two efflux systems (i.e., 6 out of 10 isolates), no genes encoding the SmeABC and SmeIJK systems were detected. In the remaining three isolates, the SmeABC and SmeYZ systems were not revealed, and in one isolate the SmeABC and SmrA systems were not detected.

### 3.3. Molecular Typing of Isolates by Pulsed-Field Gel Electrophoresis (PFGE)

Genetic relatedness between the 94 studied clinical isolates of *S. maltophilia* was assessed using PFGE analysis (Figure 1). Isolates were considered as a cluster if the similarity was at least 80%. The PFGE analysis of the obtained isolate band patterns revealed 11 clusters, named A–K, which contained at least two tested isolates. Only 41 out of the 94 studied isolates met these requirements and were mapped to clusters. The isolates included in the clusters were assigned the appropriate pulsotype number. The two groups of isolates were obtained among the largest A cluster (i.e., nine isolates created A1 PT and six isolates created A3 PT). Within each of these groups, the similarity of the isolates was 100%. These isolates were obtained from blood samples collected between September 2010 and February 2013. Moreover, only three pairs of isolates with PTs (D1, H1, and I1) obtained 100% similarity in the PFGE patterns.

### 3.4. Multilocus Sequence Typing (MLST) Analysis

The MLST analysis was performed on three selected *S. maltophilia* isolates from the most important clinical material (blood). The following representative isolates from clusters containing isolates with 100% similarity were tested: no. 56/2012 with A1 PT (formed from nine blood isolates), no. 62/2012 with A3 PT (made up of six blood isolates), and no. 57/2012 with D1 PT (containing two blood isolates). The isolates mapped in these clusters showed the presence of all MDR efflux systems tested. Information regarding the MLST data of studied isolates is presented in Table 4. The MLST analysis showed that all three investigated isolates represent two novel MLST profiles, marked as sequence types—ST498 and ST499. Isolates 56/2012 from A1 PT and 62/2012 from A3 PT had an identical allele pattern, where five out of seven alleles of the highly conserved housekeeping genes analyzed revealed new sequences. Both isolates belong to the new group ST498. The third tested isolate, 57/2012 with D1 PT, also had new gene alleles and was classified into the second new MLST profile, ST499. All new sequences of alleles of conserved housekeeping genes and new ST profiles were deposited in public databases for molecular typing (https://pubmlst.org/organisms/stenotrophomonas-maltophilia/, accessed on 19 June 2020).

## 4. Discussion

An increase of nosocomial infections caused by *S. maltophilia* strains has been observed all over the world in recent years. According to Chang et al. [27], the prevalence of *S. maltophilia* infections in the global population increased from 1.3% to 1.7% between 2007 and 2012. The isolation of these bacteria from blood samples as an etiological factor of bacteremia is of particular concern [28,29]. In this study we characterized 94 *S. maltophilia* isolates by phenotypic and genotypic methods, including 79 isolates from patients hospitalized in one tertiary Warsaw hospital, of which 27 were isolated from blood. *S. maltophilia* is an opportunistic pathogen characterized by intrinsic resistance to many different groups of antibiotics, including aminoglycosides and almost all β-lactams. Trimethoprim-sulfamethoxazole is still primarily the drug of choice for the treatment of *S. maltophilia* infections [2,30]. Screening studies conducted under the SENTRY Antimicrobial Surveillance Program, 2009–2012, showed resistance to trimethoprim-sulfamethoxazole among only 4% of *S. maltophilia* isolates from patients hospitalized with pneumonia in the United States and among 2% of isolates from European hospitals [31]. Likewise, all 94 isolates from the Warsaw hospital and from outpatients tested in this study turned out to be susceptible to trimethoprim-sulfamethoxazole. However, in the last 10 years, an increase in the number of isolates resistant to this chemotherapeutic agent has been observed in various regions of the world [28,32]. The second commonly used group of drugs in the treatment of infections caused by *S. maltophilia* strains are the fluoroquinolones, mainly levofloxacin [28,33,34]. Despite the high cure rates of *S. maltophilia* infections with levofloxacin monotherapy, a trend towards the selection of resistance to this fluoroquinolone has been observed [33]. Among the 94 *S. maltophilia* isolates tested in this study using the Etest with levofloxacin, 7% of the resistant isolates and 39% of the intermediate isolates were identified. Determination of the MIC value of antibiotics is a more accurate and reliable method of testing the drug susceptibility of bacteria than the disc-diffusion method. This is especially true specially in cases of obtaining results close to the cut-off points which determine the susceptibility of the isolate. However, some diagnostic laboratories issue results based solely on the disc-diffusion method.

The following antibiotics may also be used to treat *S. maltophilia* infections, mainly those caused by trimethoprim-sulfamethoxazole-resistant strains: minocycline, tigecycline, and the less frequently used ceftazidime and ticarcillin-clavulanate [28,35]. All 94 isolates tested in our study were susceptible to minocycline. Recently, the United States Food and Drug Administration (FDA) and the European Medicines Agency (EMA) approved a new drug, cefiderocol, which also presents high activity against *S. maltophilia* strains [36].

The main cause of difficulties in the treatment of infections caused by *S. maltophilia* is the intrinsic and acquired resistance of these strains to a wide range of antibiotics and chemotherapeutic agents. All strains produce L1 and L2 β-lactamases, which confer intrinsic resistance to β-lactam antibiotics [4]. The second important mechanism of resistance is active drug removal from bacterial cells by MDR efflux pumps. The sequencing of genomes of the *S. maltophilia* strains revealed the presence of MDR efflux systems from the RND family (*n* = 8), ABC family (*n* = 4), and MFS family (*n* = 1) [12,13,14,15,17]. The SmeDEF and SmeABC efflux systems were detected in almost all studied strains [8,37]. However, no analysis of the occurrence of so many MDR efflux pumps in such a large group of isolates as in our study has been performed so far. Recent studies of the whole-genome sequencing of 375 strains isolated from different environments (from humans, animals, and the natural environment) focused mainly on the genogroup organization and diversity of *S. maltophilia* [38]. Moreover, mobile genetic elements and some antimicrobial-resistance genes were identified. However, no in-depth analysis of the occurrence of MDR efflux pumps was performed. Only the presence of genes encoding RND efflux pump proteins as subunit AcrA, and not the more closely identified permease subunit of the transporter, was signaled [38].

We demonstrated the prevalence of all MDR efflux systems from the RND (*n* = 8) and ABC (*n* = 2) families among the majority of tested human isolates, including those derived from blood. The least-frequent MDR efflux pump in the studied isolates was SmeABC, which was detected in only 72 out of 94 isolates. It is known that contrary to MFS pumps, both RND and ABC efflux pumps have a wide substrate range, extruding trimethoprim-sulfamethoxazole, fluoroquinolones, aminoglycosides, tetracyclines, β-lactams, and chloramphenicol [4,8,37]. Overexpression of these MDR efflux systems may result in resistance or decreased susceptibility in *S. maltophilia* strains. Thus, it could be the cause of therapeutic failure. Recently, it was noted that vitamin K3 induces the expression of the *smeVWX* efflux pump genes and confers resistance to quinolones, chloramphenicol, and tetracycline [39].

Generally, *S. maltophilia* isolates exhibit high genetic diversity. Genotyping of these isolates can be performed by various methods, such as PFGE, MLST, multilocus variable number of tandem repeat analysis (MLVA), restriction fragment length polymorphism analysis (RFLP) of the *gyrB* gene, amplified fragment length polymorphism analysis (AFLP) like repetitive extragenic palindromic-PCR (Rep-PCR), and enterobacterial repetitive intergenic consensus PCR (ERIC-PCR) [18,19,24,26,40]. The PFGE method is still considered the “gold standard” in the molecular typing of bacterial isolates, including *S. maltophilia*. PFGE analysis makes it possible to prove how closely related isolates tested in the laboratory are, and whether an outbreak of infections has occurred. However, unlike PFGE, methods based on PCR and sequencing allow for the comparison of results obtained in various laboratories around the world. Recently, bacterial whole-genome sequencing (WGS) has been used in scientific and epidemiological research [20].

Genotyping by PFGE and MLST is most commonly used to determine the relationship of *S. maltophilia* isolates worldwide [18,19,24,26]. In this study, the PFGE analysis revealed a clonal relationship with 100% similarity among two sub-PTs: the first included nine bloodstream isolates derived from patients of one hospital ward, and the second included six. It should be emphasized that the presence of genes encoding both the eight pumps from the RND family and the two pumps from the ABC family was found in all these isolates. The MLST genotyping of isolates representative of these two sub-PTs showed that they belong to the same new MLST profile, named as ST498. On the other hand, in PFGE, only 81.9% similarity was observed between the two sub-PTs: A1 PT and A3 PT, despite these isolates belonging to the same MLST profile (ST498). Moreover, a correlation with 100% similarity in PFGE analysis was observed for the other three pairs of isolates. The MLST analysis revealed that one of these isolates (classified under D1 PT) belonged to another new profile, named ST499. Both new ST profiles were deposited in public databases for molecular typing. Apart from this small group of above-mentioned 21 isolates, molecular genotyping revealed no other epidemiological incidents in both groups of hospitalized patients and outpatients. Recently, some data has been published showing the transmission of *S. maltophilia* strains not only between patients but also via items such as taps and beds in hospital wards [2,28,41]. PFGE and MLST genotyping were performed on 21 clinical isolates (from blood and urine) and 9 environmental isolates of *S. maltophilia* recovered from faucets in the emergency department in a Mexican tertiary care hospital. All nine environmental isolates showed 100% similarity to clinical blood isolates and were grouped into three PTs and two STs [28]. In our study, the bloodstream isolates with 100% similarity, grouped into two sub-PTs and the same ST498, were isolated over a long period of time (i.e., 3 years). Therefore, this may indicate the spread of *S. maltophilia* from the environment to patients. It is well known that strains of *S. maltophilia* are ubiquitous in the aquatic environment [1,2,3]. Perhaps, in this case, the *S. maltophilia* strains could live in the hospital ward environment (e.g., in taps or washbasin siphons), and sometimes be transmitted to immunocompromised patients, where the infection develops. The phenomenon of infecting patients with hospital strains living in the moist environment is not limited to *Stenotrophomonas* strains. This situation was described and epidemiologically investigated in the case of other bacteria species, such as *Acinetobacter* [42], *Pseudomonas* [43], and several *Enterobacteriaceae* species [44].

## 5. Conclusions

Although a large genetic diversity is generally observed among clinical isolates of *S. maltophilia,* we noticed two small outbreaks of infections in one of the hospitals in Warsaw over a three-year period. Most likely, the strains were being transferred from the hospital environment to immunocompromised patients. Moreover, two new MLST profiles (ST498 and ST499) were discovered and described among the studied *S. maltophilia* isolates. Analysis of the susceptibility profiles showed the possibility of using the first-line drug (i.e., trimethoprim-sulfamethoxazole) as well as minocycline in the treatment of infections. On the other hand, the reduction in susceptibility to levofloxacin demonstrated in 44/94 isolates may indicate the ineffectiveness of using this fluoroquinolone in monotherapy. It should be emphasized that the presence of genes encoding eight efflux pumps from the RND family, as well as two efflux pumps from the ABC family, was shown in the majority of the 94 isolates. Such widespread occurrence of so many MDR efflux systems in clinical isolates of *S. maltophilia* indicates a high possibility of induction of the overexpression of these pumps, which would make these strains resistant to a wide spectrum of antibiotics.

## Figures and Tables

**Figure 1 biology-11-00105-f001:**
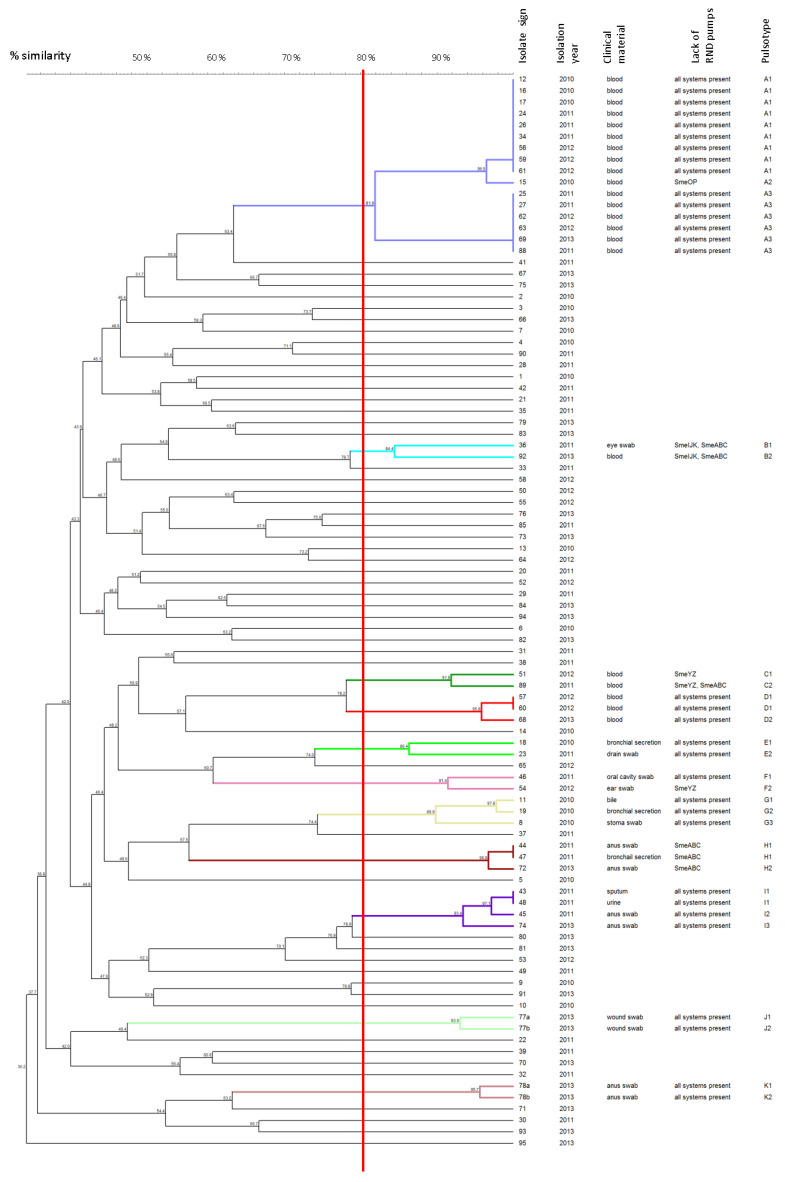
Analysis of PFGE patterns of the *S. maltophilia* isolates. The dendrogram presents the percentage similarity of PFGE profiles, the isolation year, the pulsotype mapped to clusters, and the following data about the named PTs: specified absence of the RND efflux systems and clinical material of isolation. The solid line indicates 80% similarity and is used to define the clusters with appropriate PTs. Isolates mapped to different clusters are marked with the various colors. In contrast, isolates not grouped in clusters are marked in black.

**Table 1 biology-11-00105-t001:** Primers for the amplification of genes encoding MDR efflux pumps.

Efflux System	Target Gene	Primer	Sequence (5′–3′)	Product Length (bp)	Annealing Temperature (°C)	Reference
SmeABC	*smeB*	B-F	GGGCCGGAAAGCTACGA	200	59	Chang et al. [8]
		B-R	AGCGAAATGGTCACGAATGG			
	*smeA*	A-F	AAGGCCATCGATGGCAAGGC	146	59	This study
		A-R	TCCGGGTTCGGAATGACCG			
SmeDEF	*smeD*	D-F	CCAAGAGCCTTTCCGTCAT	150	59	Zhang et al. [6]
		D-R	TCTCGGACTTCAGCGTGAC			
		RT-D-F	CGGTCAGCATCCTGATGGA	73	59	Garcia-Leon et al. [7]
		RT-D-R	ACGCTGACTTCGGAGAACTC			
SmeYZ	*smeZ*	Z-F	AGTGGACCAGCCAGTCGCT	508	59	This study
		Z-R	ACTACATAGAAGACCGGCACG			
SmeIJK	*smeK*	K-F	GACCTCGCAGACGCAGTCG	505	59	Gould et al. [9] modified
		K-R	CAGGTAGTCGCGCAGGGTC			
	*smeI*	I-F	TTCCGCGAAGGCCAGGAAGT	107	59	This study
		I-R	TCGTTCTGGCGCTTGGCTG			
SmeOP	*smeP*	P-F	GGTGCTGGCGATGACCTTC	372	58	This study
		P-R	TCCGGCAGCA TCTTGTCGC			
SmeMN	*smeN*	N-F	GGTCTCCTCG ACCATGGAC	314	58	This study
		N-R	CCTTGCCCAGCGGGATG			
SmeVWX	*smeW*	W-F	TTCGGCGACATCGTGCTCAA	843	58	This study
		W-R	CTTGAAGAAGCGGTTGAACGG			
SmeGH	*smeH*	H-F	GTGGATGATCGGCTTCACGAT	556	58	This study
		H-R	CGCATAGCCCTGGTCTTCTT			
MacABCsm	*macB*	MacB-F	GTGATCGACGAGAACACCCA	589	58	This study
		MacB-R	GGCCGATCATCGAGCCCA			
SmrA	*smrA*	SmrA-F	GGTGTGGCCGGTGCTGCT	677	63	This study
		SmrA-R	CGCGGTGCTTGACCGCCA			

F, forward primer; R, reverse primer.

**Table 2 biology-11-00105-t002:** The drug susceptibility of *S. maltophilia* clinical isolates (*n* = 94).

Agent	Disc-Diffusion Method				Etest Method					
	No. of Isolates (%)			GIZ Range ^a^ (mm)	No. of Isolates (%)			MIC Range (mg/L)	MIC_50_ (mg/L)	MIC_90_ (mg/L)
	S	I	R		S	I	R			
Minocycline	94 (100)	0	0	21–36	94 (100)	0	0	0.19–3	0.75	1.5
Levofloxacin	86 (91)	6 (6)	2 (2)	9–34	50 (53)	37 (39.5)	7 (7.5)	1–16	2	6
Trimethoprim- sulfamethoxazole	94 (100)	0	0	18–37	94 (100)	0	0	0.047–0.75	0.125	0.25

S, susceptible strain; I, intermediate strain; R, resistant strain; GIZ, growth inhibition zone of the strain; MIC, the minimum inhibitory concentration of agent. ^a^ Range of obtained diameters of the bacteria growth inhibition zones.

**Table 3 biology-11-00105-t003:** The occurrence of MDR efflux systems in *S. maltophilia* clinical isolates (*n* = 94).

Efflux System Family	Efflux System	No. of Isolates (%)
RND	SmeABC	72 (76.6)
	SmeDEF	94 (100)
	SmeIJK	86 (91.5)
	SmeYZ	89 (94.7)
	SmeMN	94 (100)
	SmeOP	90 (95.7)
	SmeVWX	93 (98.9)
	SmeGH	94 (100)
ABC	SmrA	93 (98.9)
	MacABCsm	94 (100)

**Table 4 biology-11-00105-t004:** Alleles and sequence types in *S. maltophilia* isolates.

Strain	Alleles							ST
	*recA*	*gapA*	*guaA*	*atpD*	*nuoD*	*ppsA*	*mutM*	
56/2012	74	157 ^a^	273 ^a^	130 ^a^	139	186 ^a^	148 ^a^	498 ^b^
57/2012	150 ^a^	18	274 ^a^	6	142 ^a^	185 ^a^	148 ^a^	499 ^b^
62/2012	74	157 ^a^	273 ^a^	130 ^a^	139	186 ^a^	148 ^a^	498 ^b^

^a^ New alleles. ^b^ New STs.

## Data Availability

New sequence types were submitted to the *S. maltophilia* MLST database: ST498 (https://pubmlst.org/bigsdb?page=profileInfo&db=pubmlst_smaltophilia_seqdef&scheme_id=1&profile_id=498, accessed on 19 June 2020) and ST499 (https://pubmlst.org/bigsdb?page=profileInfo&db=pubmlst_smaltophilia_seqdef&scheme_id=1&profile_id=499, accessed on 19 June 2020).

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
