# Peer review of "Phenotypic and Molecular Characteristics of the MDR Efflux Pump Gene-Carrying Stenotrophomonas maltophilia Strains Isolated in Warsaw, Poland"

_biology, 2022, doi:10.3390/biology11010105_

Round 1

Reviewer 1 Report

Review on manuscript "biology-1520413"

The manuscript to biology titled: "Phenotypic and Molecular Characteristics of the MDR Efflux Pump Gene-Carrying Stenotrophomonas maltophilia Strains Isolated in Warsaw, Poland" describes the characterization of 79 clinical isolates of Stenotrophomonas maltophilia using AMR testing, PFGE, detection of MDR efflux pumps, and in selected isolates MLST analysis. The thematic issue is of interest to the reader and is of good quality but could benefit from an additional short paragraph about clinics and epidemiology of Stenotrophomonas maltophilia in the introduction, as well as an additional part in the discussion (a little more as seen in line 339-342). 

Further small changes would be of added value:

line 16-19: rephrase sentence, it is not clear what you mean.

line 40: add "a" before "majority"

line 44: "includes" instead of "including"

line 55: delete "the" before "RND"

line 63: "of" instead of "occurred to"

line 64: use "are" instead of "were"

line 90-93: rephrase sentence, it is not clear to the reader

line 103, 113: exchange "strains" to isolates of bacterial cultures; also check throughout the document the use of strains instead of isolates

line 137: use "correct use" instead of "correctness use"

line 195-202: write names of genes in italics

line 205: write name Stenotrophomonas maltophilia in italics

line 337 & 350: full stop after i. e. 

339-342: the discussion would benefit from similar paragraphs as seen here.

Author Response

Comments and Suggestions for Authors

Review on manuscript "biology-1520413"

The manuscript to biology titled: "Phenotypic and Molecular Characteristics of the MDR Efflux Pump Gene-Carrying Stenotrophomonas maltophilia Strains Isolated in Warsaw, Poland" describes the characterization of 79 clinical isolates of Stenotrophomonas maltophilia using AMR testing, PFGE, detection of MDR efflux pumps, and in selected isolates MLST analysis. The thematic issue is of interest to the reader and is of good quality but could benefit from an additional short paragraph about clinics and epidemiology of Stenotrophomonas maltophilia in the introduction, as well as an additional part in the discussion (a little more as seen in line 339-342). 

Our answer: Corrected.

In the first paragraph of the Introduction we add the information about clinics and epidemiology of Stenotrophomonas maltophilia:

“Stenotrophomonas maltophilia is one of the most frequently isolated non-fermentative Gram-negative bacteria responsible for nosocomial infections [1-3]. This opportunistic pathogen is dangerous for patients with coexisting diseases like, acquired immunodeficiency syndrome (AIDS), cystic fibrosis,  or cancer (particularly lung cancer), as well as patients with other immunodeficiencies, immunosuppressive therapy, mechanical ventilation, catheters, and those hospitalized for long periods. The critically ill patients from intensive care units, especially after an broad-spectrum antibiotic therapy, are most exposed to infections. S. maltophilia strains cause mostly nosocomial respiratory tract infections (pneumonia) which can be associated with bloodstream infections. This bacteria can cause many other severe infections, including respiratory infections associated with acute exacerbations of chronic obstructive pulmonary disease, eye infections (endophthalmitis, keratitis, scleritis), skin and soft tissue infections, meningitis, endocarditis, and biliary sepsis. Infections caused by S. maltophilia are characterized by a high mortality rate, even up to 69% in patients with bacteraemia [3]. Although it is mainly a nosocomial pathogen, the community-acquired infections are increasingly observed. Infections caused by this bacterium occur in both adults and children [1,2]. Infection occurs most often through direct contact with the aerosol created by a person with pneumonia. Other possible routes of transmission involve, the hands of healthcare workers, dental unit suction tubing, contaminated endoscopes, or tap water. S. maltiphilia strains, like most non-fermentative Gram-negative bacteria, are common in wet or moist environments, e.g. in taps or washbasin siphons and their surroundings [2].”

At the end of the discussion, we added a sentence:

The phenomenon of infecting patients with hospital strains living in the moist environment is not limited  only to the Stenotrophomonas strains. This situation was described and epidemiologically investigated in the case of other bacteria species like, Acinetobacter [42], Pseudomonas [43] and several Enterobacteriaceae species [44].

Further small changes would be of added value:

line 16-19: rephrase sentence, it is not clear what you mean.

Our answer: Corrected.

The sentence in the text “Although a large genetic variation was observed among the clinical strains of S. maltophilia, the pulsed field gel electrophoresis analysis revealed a clonal relationship, with 100% similarity among two sub-pulsotypes: the first included nine bloodstream isolates and the second involved six.”

was changed to a sentence

“A large genetic variation was observed among the isolates tested by the pulsed field gel electrophoresis. A clonal relationship, with 100% similarity, was observed between isolates within two sub-pulsotypes, respectively: the first included nine bloodstream isolates and the second involved six.”

line 40: add "a" before "majority"

Our answer: Corrected.

line 44: "includes" instead of "including"

Our answer: Corrected.

line 55: delete "the" before "RND"

Our answer: Corrected.

line 63: "of" instead of "occurred to"

Our answer: Corrected.

line 64: use "are" instead of "were"

Our answer: Corrected.

line 90-93: rephrase sentence, it is not clear to the reader

Our answer: Corrected.

The sentence in the text “However, the knowledge obtained about the prevalence of efflux pump genes in majority of clinical S. maltophilia strains makes it possible to predict the scale of the risk of resistance emergence in strains as a result of gene overexpression.”

was changed to a sentence

“So, if the majority of the clinical S. maltophilia  strains possess the several above genes, the scale of the existing danger of the emergence of resistance, even MDR, is huge.”

line 103, 113: exchange "strains" to isolates of bacterial cultures; also check throughout the document the use of strains instead of isolates

Our answer: Corrected.

line 137: use "correct use" instead of "correctness use"

Our answer: Corrected.

line 195-202: write names of genes in italics

Our answer: Corrected.

line 205: write name Stenotrophomonas maltophilia in italics

Our answer: Corrected.

line 337 & 350: full stop after i. e. 

Our answer: Corrected.

339-342: the discussion would benefit from similar paragraphs as seen here.

Our answer: Corrected.

At the end of the discussion, we added a sentence:

The phenomenon of infecting patients with hospital strains living in the moist environment is not limited  only to the Stenotrophomonas strains. This situation was described and epidemiologically investigated in the case of other bacteria species like, Acinetobacter [41], Pseudomonas [42] and several Enterobacteriaceae species [43].

Submission Date 10 December 2021

Date of this review 21 Dec 2021 17:49:21

Reviewer 2 Report

Olga et al., in their article titled “Phenotypic and Molecular Characteristics of the MDR Efflux 2 Pump Gene-Carrying Stenotrophomonas maltophilia Strains 3 Isolated in Warsaw, Poland”,has been well summarized, however, there are a few minor issues with the manuscript as mentioned below

  • Line 14 states the source of 79 isolates of 94. What is the source of the remaining 15 isolates?

Is it essential to state the source of isolates in the abstract?

  • Line 32 maybe use of term gram negative bacteria instead of rods would be more apt.

  • Line 54-55, if possible, could be re framed like “of these the RND efflux system majorly contributes to the building of antibiotic/ antimicrobial resistance.

  • Line 63 please change the wording “occurred into” to present within

  • Line 75 please change the wording “research is ongoing on” like current ongoing research on …

  • Line 95 states various clinical materials. Please state few.

  • The fluoroquinolone antibiotics include ciprofloxacin (Cipro), gemifloxacin (Factive), levofloxacin (Levaquin), moxifloxacin (Avelox), and ofloxacin (Floxin). The authors only used three different class of fluoroquinolones for this study. Why select specifically only three? Can there be any background study or hypothesis for using only three mentioned ones?

  • Can the authors use variable tandem repeat typing in order to distinguish between ST498, ST499 and other reference strains? Multilocus VNTR analysis (MLVA) is a method which determines the number of tandem repeat sequences at different loci in a bacterial genome. Then to further distinguish the isolates the authors could setup a simple MLVA assay where in number of well-selected VNTR loci are amplified by multiplex PCR and an analysis of the amplicons is conducted on standard agarose gels.

Author Response

Comments and Suggestions for Authors

Olga et al., in their article titled “Phenotypic and Molecular Characteristics of the MDR Efflux 2 Pump Gene-Carrying Stenotrophomonas maltophilia Strains 3 Isolated in Warsaw, Poland”,has been well summarized, however, there are a few minor issues with the manuscript as mentioned below

  • Line 14 states the source of 79 isolates of 94. What is the source of the remaining 15 isolates?

Is it essential to state the source of isolates in the abstract?

Our answer: Corrected.

The sentence in the text “The phenotypic and genotypic characterization of 94 S. maltophilia isolates was performed, including 79 isolates from patients hospitalized in tertiary Warsaw hospital.”

was changed to a sentence

“The phenotypic and genotypic characterization of 94 S. maltophilia isolates was performed, including isolates from patients hospitalized in tertiary Warsaw hospital (n=79) and from outpatients (n=15).”

  • Line 32 maybe use of term gram negative bacteria instead of rods would be more apt.

Our answer: Corrected.

  • Line 54-55, if possible, could be re framed like “of these the RND efflux system majorly contributes to the building of antibiotic/ antimicrobial resistance.

Our answer: Corrected.

“Of these families, mainly RND efflux systems, contribute to antimicrobial resistance.”

  • Line 63 please change the wording “occurred into” to present within

Our answer: Corrected.

  • Line 75 please change the wording “research is ongoing on” like current ongoing research on

Our answer: Corrected.

  • Line 95 states various clinical materials. Please state few.

Our comments for Reviewer:

In the sentence regarding the purpose of the study, only blood was intentionally mentioned as the most important clinical material. In the world, the highest mortality was demonstrated among patients with bacteremia. Of course, all clinical materials from which S. maltophilia was isolated are presented below in the section 2.1 Bacterial Strains.

  • The fluoroquinolone antibiotics include ciprofloxacin (Cipro), gemifloxacin (Factive), levofloxacin (Levaquin), moxifloxacin (Avelox), and ofloxacin (Floxin). The authors only used three different class of fluoroquinolones for this study. Why select specifically only three? Can there be any background study or hypothesis for using only three mentioned ones?

Our comments for Reviewer: In the Introduction and Discussion sections, we explain why the susceptibility of the S. maltophilia isolate collection to only three antibiotics was determined, i.e. trimethoprim-sulfamethoxazole, levofloxacin and minocycline.

Lines 40-45 (now 52-57 in new clean version) “S. maltophilia infections are difficult to treat, because they are intrinsically resistant to majority of antibiotics and chemotherapeutic agents. According to the Clinical and Laboratory Standards Institute (CLSI) recommendations for Group A antimicrobial agents which are appropriate for use in a routine treatment of S. maltophilia infection, including only three the following antibiotics: trimethoprim-sulfamethoxazole, levofloxacin and minocycline.”

Lines 260-261 (now 271-272 in new clean version)  “Trimethoprim-sulfamethoxazole is still primarily the drug of choice for the treatment of S. maltophilia infections [2,30].”

Lines 268-270 (now 279-281 in new clean version)  “The second commonly used group of drugs in the treatment of infections caused by S. maltophilia strains are fluoroquinolones, mainly levofloxacin [28,33,34].”

Line 279-285 (now 290-296 in new clean version)  “Besides, the following antibiotics may be used to treat S. maltophilia infections, mainly caused by trimethoprim-sulfamethoxazole-resistant strains: minocycline, tigecycline, and the less frequently used ceftazidime and ticarcillin-clavulanate [28,35]. All 94 isolates tested in our study turned out to be susceptible to minocycline. Recently, the United States Food and Drug Administration (FDA) and the European Medicines Agency (EMA) approved a new drug, cefiderocol, which also presents high activity against S. maltophilia strains [36].”

Thus, the fluoroquinolones are the second most important group of chemotherapeutic agents after trimethoprim-sulfamethoxazole in the treatment of S. maltophilia infections in the world. The most commonly used drug from fluoroquinolone group is levofloxacin.  Nonetheless, a comparison of data from worldwide SENTRY studies reveals a decrease in sensitivity of S.maltophilia to levofloxacin. On the other hand, a few reports showed a low MIC50 (0.5 mg/L and 0.5 mg/L) and low MIC90 values (8 and 4 mg/L) for moxifloxacin against S. maltophilia, pointing to that moxifloxacin can be considered an effective alternative. Moreover, data from multiple studies show that ciprofloxacin has weak activity against S. maltophilia. Nevertheless, the clinical efficacy of fluoroquinolones other than levofloxacin still needs to be validated due to the limited number of clinical studies reported and the current lack of clinical breakpoints [Front. Microbiol. 2015, 6:893, doi: 10.3389/fmicb.2015.00893] .

Most importantly, the interpretive breakpoints for drug susceptibility are available only for levofloxacin out of fluoroquinolones in the CLSI recommendations [CLSI M100-ED31:2021 Performance Standards for Antimicrobial Susceptibility Testing, 31st Edition1]. The breakpoint for susceptibility to levofloxacin are MIC ≤ 2 mg/L. Therefore, when determining the sensitivity profiles of the strain collections in the submitted manuscript, we only examined the levofloxacin susceptibility and interpreted the results according to the CLSI recommendations.

  • Can the authors use variable tandem repeat typing in order to distinguish between ST498, ST499 and other reference strains? Multilocus VNTR analysis (MLVA) is a method which determines the number of tandem repeat sequences at different loci in a bacterial genome. Then to further distinguish the isolates the authors could setup a simple MLVA assay where in number of well-selected VNTR loci are amplified by multiplex PCR and an analysis of the amplicons is conducted on standard agarose gels.

Our answer: In the Discussion section, we have added sentences concerning the methods used to genotype S. maltophilia isolates.

Lines 325-336 in new clean version “Generally S. maltophilia isolates exhibit high genetic diversity. Genotyping these isolates can be performed by various methods as PFGE, MLST, multilocus variable number of tandem repeat analysis (MLVA), restriction fragment length polymorphism analysis (RFLP) of the gyrB gene, amplified fragment length polymorphism analysis (AFLP) like repetitive extragenic palindromic-PCR (Rep-PCR) and enterobacterial repetitive intergenic consensus PCR (ERIC-PCR) [18,19,24,26,40]. Until now, the PFGE method is still considered the “gold standard” in molecular typing of bacterial isolates, including S. maltophilia. The PFGE analysis allows to prove how closely related are the isolates tested in the laboratory, and whether an outbreak of infections has occurred. However, unlike PFGE, methods based on PCR and sequencing allow for the comparison of results obtained in various laboratories around the world. Recently, the bacterial whole genome sequencing (WGS) has been used in scientific and epidemiological  research [20].”

Our comments for Reviewer: The MLVA analysis for genotyping S. maltophilia isolates was first described in 2008 by Roscetto et al. [BMC Microbiol. 2008, 8:202]. Molecular typing of 38 S. maltophilia isolates, including the K279a strain with a known sequenced genome, was performed using both the MLST and PFGE methods. Overall, the MLVA data matched the genotyping data obtained by PFGE. However, some isolates presenting the same PCR profiles in all loci showed different PFGE standards. To date, PFGE, right after WGS, is the most accurate method of studying the relation between isolates in one laboratory.

So, to understand how are closely related all 94 S. maltophilia isolates, they were genotyped using PFGE during research described  in the submitted manuscript.

Submission Date

10 December 2021

Date of this review

23 Dec 2021 20:57:58
